# Wafer-Scale Room-Temperature Bonding of Smooth Au/Ti-Based Getter Layer for Vacuum Packaging

**DOI:** 10.3390/s22218144

**Published:** 2022-10-24

**Authors:** Takashi Matsumae, Shingo Kariya, Yuichi Kurashima, Le Hac Huong Thu, Eiji Higurashi, Masanori Hayase, Hideki Takagi

**Affiliations:** 1Device Technology Research Institute, National Institute of Advanced Industrial Science and Technology, Ibaraki 305-8564, Japan; 2Graduate School of Science and Technology, Tokyo University of Science, Chiba 278-8510, Japan; 3Graduate School of Engineering, Tohoku University, Miyagi 980-8579, Japan

**Keywords:** wafer-scale packaging, room-temperature bonding, getter layer, template stripping

## Abstract

This study demonstrates room-temperature bonding using a getter layer for the vacuum packaging of microsystems. A thick Ti layer covered with an Au layer is utilized as a getter layer because it can absorb gas molecules in the package. Additionally, smooth Au surfaces can form direct bonds for hermetic sealing at room temperature. Direct bonding using a getter layer can simplify the vacuum packaging process; however, typical getter layers are rough in bonding formation. This study demonstrates two fabrication techniques for smooth getter layers. In the first approach, the Au/Ti layer is bonded to an Au layer on a smooth SiO_2_ template, and the Au/SiO_2_ interface is mechanically exfoliated. Although the root-mean-square roughness was reduced from 2.00 to 0.98 nm, the surface was still extremely rough for direct bonding. In the second approach, an Au/Ti/Au multilayer on a smooth SiO_2_ template is bonded with a packaging substrate, and the Au/SiO_2_ interface is exfoliated. The transferred Au/Ti/Au getter layer has a smooth surface with the root-mean-square roughness of 0.54 nm and could form wafer-scale direct bonding at room temperature. We believe that the second approach would allow a simple packaging process using direct bonding of the getter layer.

## 1. Introduction

Several types of micro-electro-mechanical systems (MEMS) are degraded by gas molecules in packages. Thus, the vacuum packaging process is an important step in the performance and lifetime of MEMS devices. Low-temperature bonding techniques have been extensively studied for the packaging of MEMS assemblies to suppress thermal damage [1,2,3,4,5,6,7]. Particularly, atomically smooth Au surfaces can form atomic bonds at room temperature [8,9,10], which enables the integration of dissimilar materials regardless of the mismatches in the coefficient of thermal expansion. It has been demonstrated that wafers coated with Au/Ti (from top to bottom) bonding layers can form vacuum packaging at room temperature [11]. In these previous studies [8,9,11], the Ti layer was deposited to form a strong adhesion between the Au bonding layer and MEMS substrate.

Additionally, it is known that the Au/Ti layer can function as a non-evaporative getter (NEG), which absorbs residual gas molecules in the package [12]. When the Au/Ti layer was annealed, the thermally diffused Ti atoms reached the surface and reacted with the residual gas. We demonstrated that smooth Au/Ti films can form direct bonds at room temperature and absorb residual gas molecules by post-bonding annealing at 200 °C [13]. It is believed that the direct bonding of the Au/Ti films can contribute to a simple MEMS packaging process in which a wafer coated with the NEG film can form a seal and cause gettering, as illustrated in Figure 1. In addition, the direct bonding of MEMS device wafers having NEG films with cap wafers also allows a simple vacuum packaging process.

Studies on direct bonding typically utilize thin Au/Ti layers, such as sputter-deposited Au and Ti layers with thicknesses of 15 and 3 nm, respectively [11]. The previous studies also revealed that thick Au/Ti films are undesirable for direct bonding [11] because the surface roughness of Au [11] and Ti [14] layers increases with the increase in the film thickness by the grain growth during deposition. Hermetic sealing requires a smooth surface with a root-mean-square (RMS) roughness of 0.5 nm or less [15,16]. However, such thin Au/Ti layers cannot absorb large amounts of gaseous molecules. To study the gettering process [12], thicker Ti layers were employed; for instance, Ming Wu employed Au and Ti layers with thicknesses in the range of 5–20 and 100 nm, respectively [10]. To address this dilemma, the present study developed fabrication techniques for an Au/Ti-based getter layer with a smooth Au surface and a thick Ti layer.

One possible solution is the use of chemical mechanical polishing (CMP) on a thick Au/Ti film; however, a chemically stable Au surface has difficulty in CMP planarization [17]. Another approach is template stripping (TS). The Au film was deposited on an ultra-flat template (i.e., glass, oxidized Si substrate) and mechanically exfoliated from the template to obtain a smooth Au surface [18,19,20,21,22,23]. Our previous study demonstrated the TS process of a 2-mm-square Au/Ti-metallized chip and achieved room-temperature bonding using a smooth surface [19]. However, this size is insufficient for packaging because wafer-scale bonding is preferable in the MEMS fabrication process [24]. In the present study, we fabricated getter films with a smooth Au surface and a 100-nm-thick Ti layer using two TS-based methods. The smoothness of the Au surface and bonding formation at room temperature were evaluated for the MEMS packaging process, as illustrated in Figure 1.

## 2. Materials and Methods

In the present study, smooth getter films were fabricated by designing a TS process, and the smooth film was directly bonded to another substrate at room temperature. The Au and Ti layers were deposited using a sputter deposition machine (SME-200E, ULVAC). The metal films were formed on oxidized 4-inch-diameter Si substrates; a 300-nm-thick SiO_2_ layer was fabricated on the Si substrates in a thermal oxidation furnace. This is because the oxidized Si substrate has an atomically smooth surface and the Au film badly adheres to the SiO_2_ surface. Thus, a smooth Au surface can be fabricated via mechanical exfoliation at the Au/SiO_2_ interface. The fabricated smooth getter film was directly bonded to an Au/Ti-metalized Si substrate to demonstrate the packaging process. Atomic bonds between the Au surfaces were generated using a surface-activated bonding method; the Au surfaces were cleaned with Ar plasma and thereafter contacted with each other at room temperature. The plasma was activated at 200 W for 30 s in pure Ar gas at a pressure of 60 Pa. After the bonding step, a 100-µm-thick razor blade was inserted between the bonded wafers. This causes the exfoliation of wafers at the Au/SiO_2_ interface.

The surface morphology of the fabricated getter film was evaluated using atomic force microscopy (AFM). The quality of the direct bonding of the smooth surface to another substrate was evaluated by scanning acoustic microscopy (SAM) and a tensile tester. Additionally, the nanostructure of the bonding interface was observed using transmission electron microscopy (TEM).

## 3. Results

In this study, two TS-based processes for the fabrication of smooth getter films were demonstrated for wafer-level packaging via room-temperature direct bonding.

### 3.1. Smoothening Au/Ti Getter Film by Conventional TS Technique

The first approach utilizes a traditional TS process, as shown in Figure 2. A 100-nm-thick Ti layer was deposited on the SiO_2_ template; subsequently, a 6-nm-thick Au layer covered the Ti layer. The Au/Ti getter layer had a relatively rough surface for the formation of direct bonding. The Au/Ti layer was bonded with a 20-nm-thick Au thin film deposited on a SiO_2_ template, which has an atomically smooth surface with an RMS roughness of ~0.2 nm, at 200 °C under a bonding load of 1.27 MPa (10 kN for a 4-inch diameter). Because the deposited Au layer badly adhered to the SiO_2_ surface, the Au/SiO_2_ interface could be exfoliated without significant mechanical deformation.

Figure 3 shows a photograph of the wafers after the TS. The photograph of the thermally-oxidized Si substrate (template) is shown in the Appendix A. The appearance of the bare template substrate and that after the TS step is similar. It indicates that most of the Au film on the template substrate was transferred to the packaging substrate. Au films were present on the template substrate after the TS process because bonding was not formed in these areas owing to the wafer shape of the edges and particles adhered to the surfaces.

The surface roughness of the fabricated films was evaluated using AFM. Figure 4a–c shows the surface images of the Au layer on the template, as-deposited thick Au/Ti layer, and Au/Ti layer after the TS process, respectively. The thin Au layer was atomically smooth because the RMS roughness was approximately 0.548 ± 0.02 nm. Meanwhile, the RMS roughness of the thick Au/Ti film was 2.00 ± 0.19 nm, which is extremely high for the formation of direct bonding. After TS, the RMS roughness was reduced to 0.98 ± 0.07 nm. Although the TS process successfully decreased the roughness of the getter layer, the RMS roughness was still extremely high for hermetic sealing [15,16].

### 3.2. Transfering Au/Ti/Au Getter Layer from Template to Packaging Substrate

Another approach is to transfer the Au/Ti/Au getter layer fabricated on the SiO_2_ template to the packaging substrate, as illustrated in Figure 5. Unlike the conventional TS process, which smoothens the surface of the thick Au/Ti layer, it is expected that the bonding surface can have a morphology similar to that of the smooth template and is less affected by the increased roughness of the thick layers. First, Au/Ti/Au getter films with thicknesses of 20/100/20 nm were deposited on thermally oxidized Si substrates. This film was thereafter bonded to an Au/Ti bonding film with a thickness of 20/5 nm on the packaging substrate. Finally, the Au/SiO_2_ interface was mechanically exfoliated.

Figure 6 shows a photograph of the wafers after the transfer process illustrated in Figure 5. Similar to the previous approach, the Au/Ti/Au film was also successfully transferred to the packaging wafer, except for the edges and area where the particles adhered.

AFM images of the Au/Ti/Au getter film before and after the transfer process are shown in Figure 7. The surface of the Au/Ti/Au layer on the SiO_2_ template was rough because the RMS roughness was 3.48 ± 0.09 nm owing to grain growth during the deposition step. Meanwhile, the Au/Ti layer was atomically thin; the RMS roughness was 0.548 ± 0.01 nm. The surface of the transferred Au/Ti/Au getter layer is shown in Figure 7c. The RMS roughness was 0.538 ± 0.12 nm, which is similar to that of the thin film. This is possibly because the surface exfoliated from the template hardly suffered from the increase in the roughness of the thick layer.

To demonstrate the packaging process using the transferred Au/Ti/Au getter layer, the surface was bonded with another SiO_2_/Si substrate with a 20/5-nm-thick Au/Ti bonding layer at room temperature, as illustrated in Figure 8. The Au surfaces were cleaned by irradiation with Ar plasma and thereafter contacted at room temperature under a bonding load of 123 kPa (1000 N) in a vacuum of 1 × 10^−2^ Pa.

The room-temperature bonding of the transferred getter layer was evaluated by SAM. Figure 9 shows a void-mapping image of the bonded substrates. The white area indicates that the ultrasonic wave was reflected by the gap at the bonding interface. Some voids coincided with the areas where the Au/Ti/Au film was not transferred, as shown in Figure 6. Additionally, other voids were presumably caused by the new particles adhering to the surfaces after the transfer process. The ratio of unbonded areas is 9.5%; it is expected that these unbonded areas will possibly disappear when the cleanliness of the process environment is improved.

The mechanical strength of the bonded substrates shown in Figure 8 was evaluated using a tensile tester. The bonded Si substrates were diced into 10-mm-square chips and glued to jigs for the tensile test. The bonded substrates did not fracture when the tensile force reached 2600 N, which is the maximum value of the tensile tester. The calculated tensile strength was greater than 26 MPa, which is significantly high for MEMS packaging. It is known that the interfacial strength over 10 MPa is equivalent to the bulk strength of Si substrates.

The interface formed in the room-temperature bonding experiment was observed using SEM and TEM. The cross-sectional surfaces of the diced chips were smoothened using an ion polisher. Figure 10 shows the SEM images of the bonded films. Two bonding interfaces were formed by the first bonding step of transferring the getter layer (Figure 5) and the second bonding step for packaging (Figure 8). In the SEM image, some large unbonded regions were observed as dark areas at the Au/Au interface formed during the first bonding step. This is because the surface of the deposited Au/Ti/Au film was rough (RMS: 3.48 ± 0.09 nm) and thus did not contact well at the first bonding step. Meanwhile, gaps were hardly observed at the second bonding interface because the Au surface after the transfer step was significantly smooth (RMS: 0.538 ± 0.12 nm).

Nanoscale observations were performed using TEM, as shown in Figure 11. The ultrathin TEM specimen was prepared using a focused ion beam on the cross-sectional surface of the diced chip. In this image, the unbonded areas at the Au/Au bonding interfaces were observed as brighter regions. At the interface formed by the first bonding step, the unbonded areas tended to be larger, and some were connected to each other. The connected unbonded areas may become leak paths to the packaged environment. Meanwhile, as shown in the bottom image, the size of the unbonded areas is 10 nm or less, and these voids are not connected. Thus, it is assumed that the leak path was not formed by the direct bonding of the transferred Au/Ti/Au getter layer. In our previous studies, the Au/Au bonding interface with such voids could form hermetic sealing at room temperature [25,26]. In addition, the hermeticity was improved by the following annealing step for getter activation [26].

## 4. Discussion

In this study, the thickness of the Ti layer was adjusted to 100 nm, which is the same as that in a previous study on the gettering process [12]. However, in the industry, thicker Ti layers are sometimes used to getter large amounts of gas molecules. In this case, the roughness of the getter layer increased further. The AFM results indicated that the conventional TS approach could reduce the RMS surface roughness from 2.00 ± 0.19 to 0.98 ± 0.07 nm. Nevertheless, the reduction was insufficient, and it was expected that bonding would be difficult in the case of a thick Ti layer. Meanwhile, by transferring the getter layer, an atomically smooth surface was obtained. As shown in the AFM images in Figure 7, the RMS roughness of the getter layer was 0.538 ± 0.12 nm, which is similar to that of the thin Au/Ti layer (0.548 ± 0.01 nm). Because the second approach enables the exfoliation of the atomically smooth Au surface from the template, the quality of the packaging is presumably less affected by the increased roughness of the thick film. Thus, we believe that the approach using the transfer technique would contribute to the vacuum packaging process using direct bonding of the getter layers.

## 5. Conclusions

In this study, an Au/Ti-based getter film was directly bonded to another substrate at room temperature for a simplified vacuum packaging process. Because hermetic sealing requires the bonding of atomically smooth Au surfaces, the fabrication processes of the smooth getter film were designed based on a TS technique. In the first approach, an Au/Ti getter layer with a thickness of 6/100 nm was deposited on a packaging substrate. The RMS roughness of the Au/Ti film was initially 2.00 ± 0.19 nm. This was bonded to a 20-nm-thick Au layer on a smooth SiO_2_ template, and the Au/SiO_2_ interface was mechanically exfoliated. Although this process reduced the surface roughness to 0.98 ± 0.07 nm, this value is undesirable for vacuum packaging. In the second approach, a 20/100/20-nm thick Au/Ti/Au getter layer was formed on the SiO_2_ template. This layer was transferred to the packaging substrate with an Au/Ti bonding layer, which was bonded to the Au/Ti bonding layer, and the smooth Au/SiO_2_ interface was exfoliated. The surface roughness of the transferred getter film was significantly reduced because the RMS surface roughness was 0.538 ± 0.12 nm. The results of the bonding experiments indicated that the transferred Au/Ti/Au getter layer could form wafer-scale direct bonding at room temperature; observations using electron microscopes suggests that ~10-nm-diameter voids were present at the bonding interface but not connected with others. Thus, it is expected that hermetic sealing without a leak path can be fabricated by the direct bonding of the transferred getter layers. We believe that this approach would allow for a simple packaging process for microsystems.

## Figures and Tables

**Figure 1 sensors-22-08144-f001:**
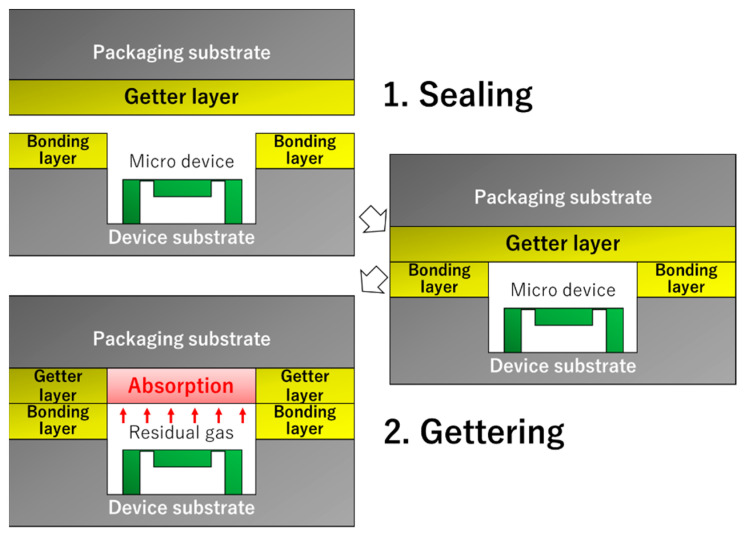
Simplified vacuum packaging process using direct bonding of getter layer.

**Figure 2 sensors-22-08144-f002:**
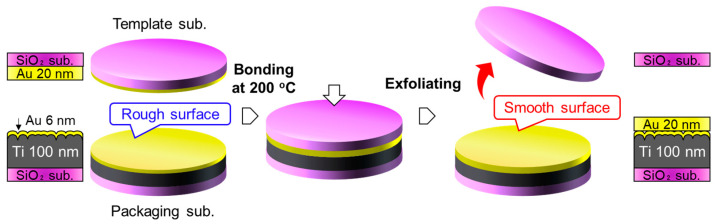
TS process for fabrication of smooth getter layer on packaging substrate.

**Figure 3 sensors-22-08144-f003:**
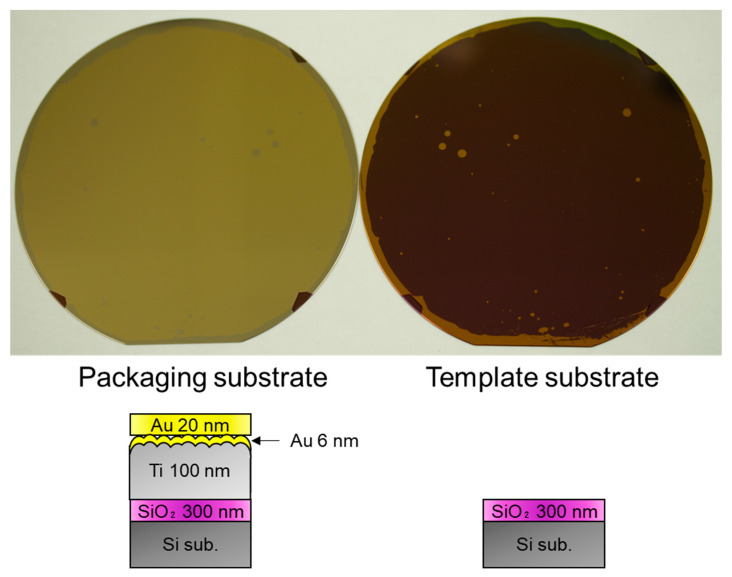
Getter film and template substrate after TS step.

**Figure 4 sensors-22-08144-f004:**
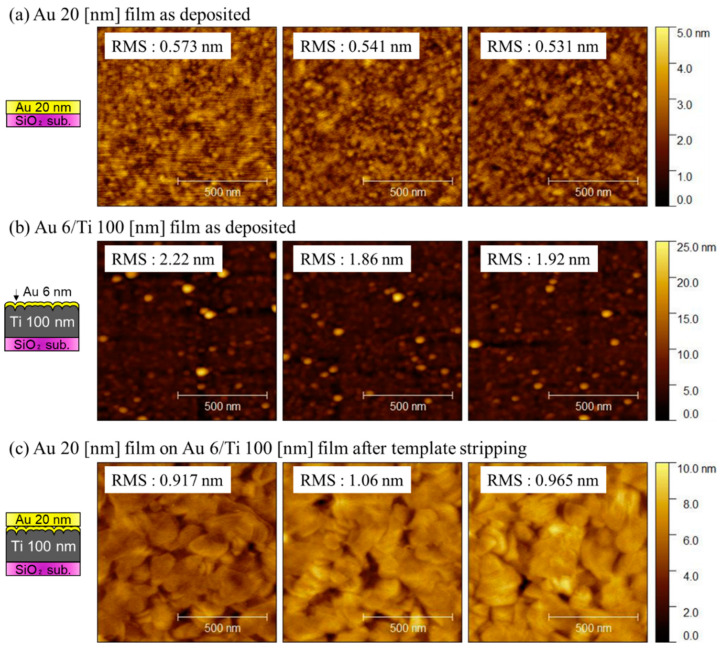
AFM images of the (**a**) thin Au layer on the SiO_2_ template, (**b**) thick Au/Ti getter, and (**c**) Au/Ti getter layer smoothened by the TS process.

**Figure 5 sensors-22-08144-f005:**
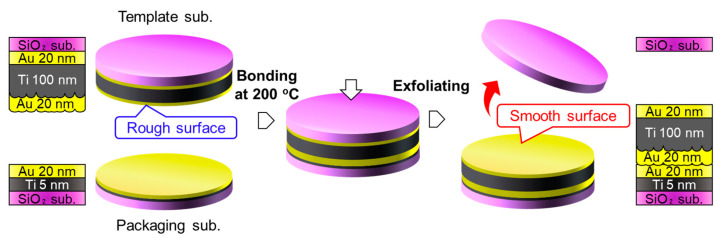
Second approach to transfer smooth Au/Ti/Au getter layer to packaging substrate.

**Figure 6 sensors-22-08144-f006:**
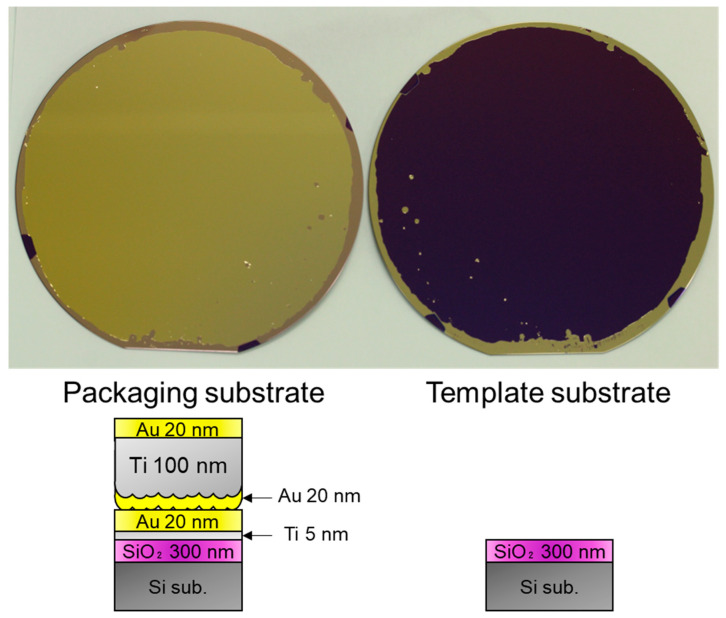
Packaging and template wafers after transferring the Au/Ti/Au layer.

**Figure 7 sensors-22-08144-f007:**
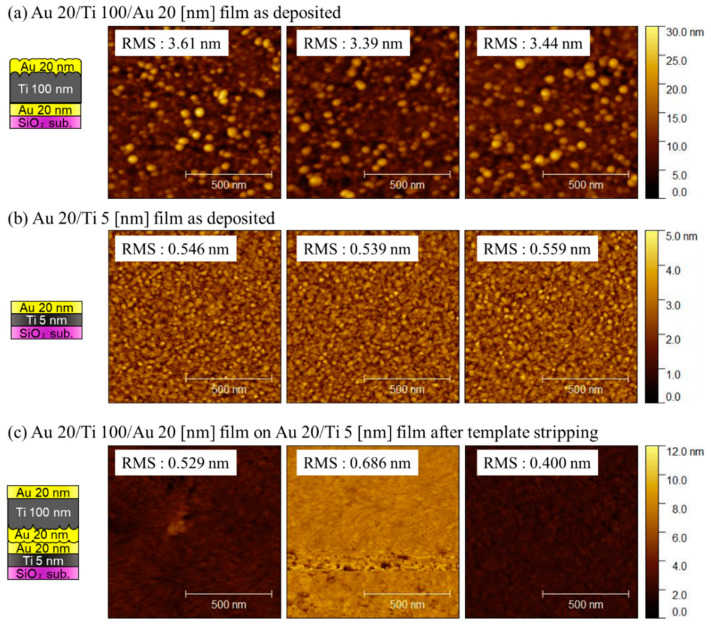
AFM images of (**a**) Au/Ti/Au getter layer on the SiO_2_ template, (**b**) thin Au/Ti bonding layer, and (**c**) transferred Au/Ti/Au getter layer on the packaging substrate.

**Figure 8 sensors-22-08144-f008:**
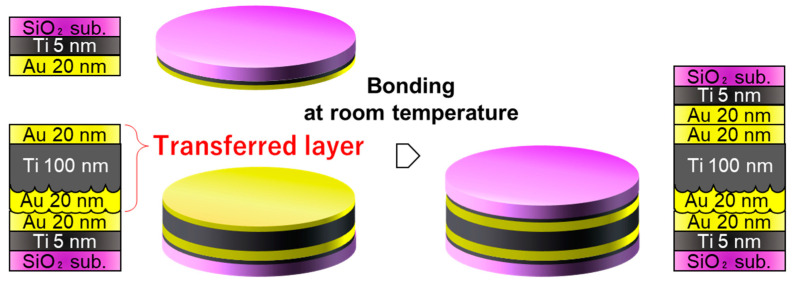
Room temperature bonding of transferred getter layer. They were bonded in a vacuum after Ar plasma irradiation.

**Figure 9 sensors-22-08144-f009:**
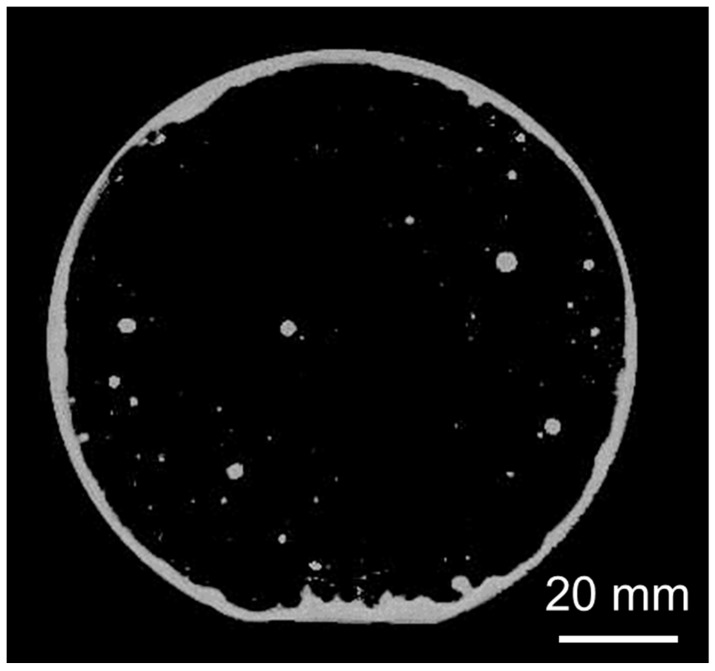
SAM image of Si wafers bonded using the transferred Au/Ti/Au getter layer and the Au/Ti bonding layer.

**Figure 10 sensors-22-08144-f010:**
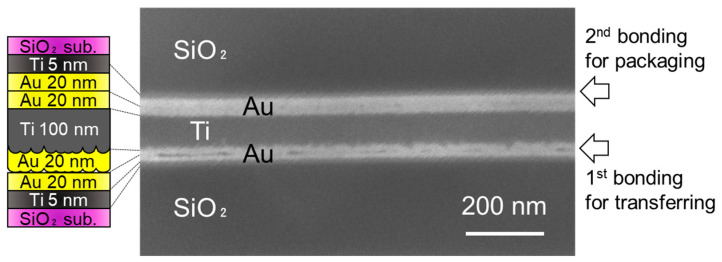
SEM image of the bonding interface. Sub-micro gaps were observed at the first bonding interface but not at the second one.

**Figure 11 sensors-22-08144-f011:**
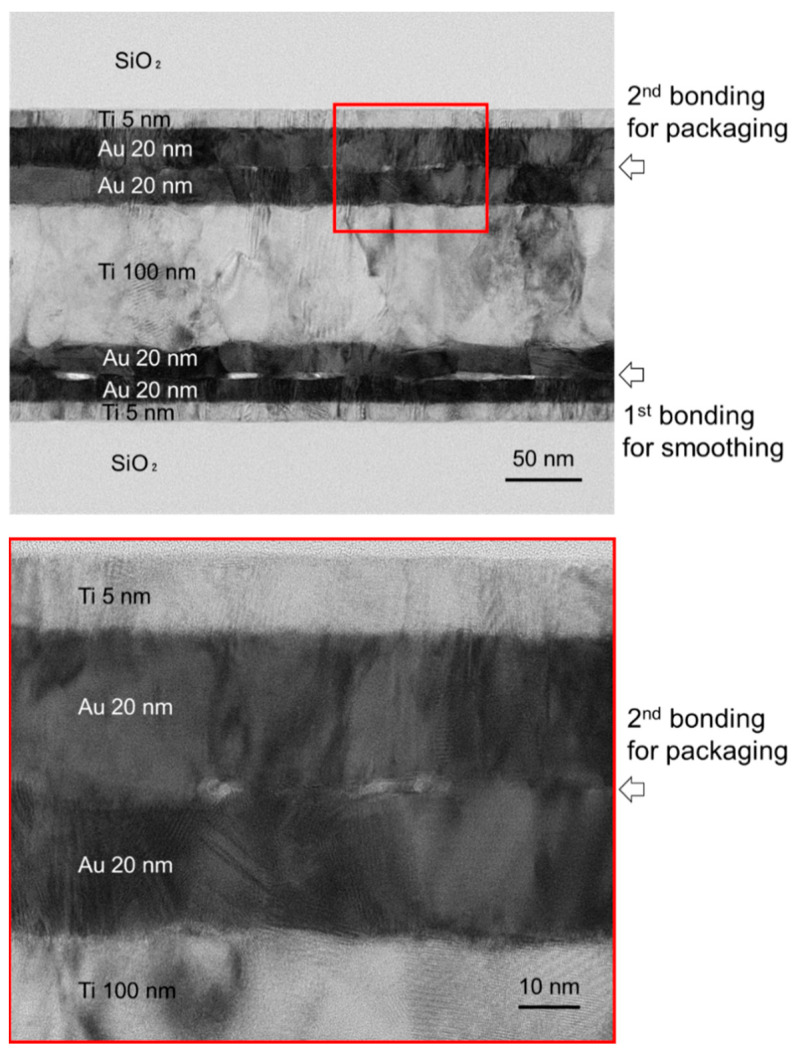
TEM image of the bonding interface. Although nanovoids were observed at the bonding interface for packaging, they did not tend to connect with others.

## Data Availability

The data presented in this study are available on request from the corresponding author.

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
