# Peer review of "Wafer-Scale Room-Temperature Bonding of Smooth Au/Ti-Based Getter Layer for Vacuum Packaging"

_sensors, 2022, doi:10.3390/s22218144_

Round 1

Reviewer 1 Report

The authors present a clever method of preparing a Au/Ti layer that is both thick enough for gettering and smooth enough for room-temp direct bonding. The manuscript is presented logically and concise. The authors do a fine job analysing the samples and presenting the found results.

The improvement in smoothness and the achieved bond strength is convincing, however there are a few major issues as well as improvements for clarity of the manuscript listed below.

major issues:

- It's a real shame the authors did not perform any experiments on hermeticity (e.g. bonding to a simple device substrate with cavities and checking the leak rate by residual gas analysis). Without it, this manuscript is only about a bonding step of two flat Au surfaces, which much less impactful.

In it's current form, the manuscript does not prove whether this bonding method is hermetic or not.

- An analysis of the bond interface, bond strength, voids, and preferably also hermeticity after a getter activation annealing is missing. The authors only show these analyses after bonding, without annealing at 200°C (or 300 °C as stated in the authors' previous paper). This is important since a real device packaged this way would spend most of its life after annealing.

- Almost 1/2 of all citations are self-citations, or associated citations. Please include more research from other authors, especially about hermetic metal-metal bonding, and about atomically flat metal surfaces.

- The authors don't address the issue of how to achieve atomically flat Au surfaces on a real device wafer. How feasible is this method in reality?

Points to improve clarity:

- Please explain how exactly the exfoliation process was performed.

- Figure 2,5,8,etc: the colour of the SiO2 substrate could be confused with a pure SiO2 material, rather than an oxidised Si wafer. Please indicate the oxidised Si wafer more clearly.

- In several places, "grain growth" or "growth surface" are used to explain the roughness of the exfoliated Au surface. This is very unclear. Does it refer to the deposition process, or the bonding step at 200°C? Please explain this better.

- Was the tensile bond strength test performed on both Fig 5 and Fig 8 samples, or only Fig 8?

- p11, line 200: Please provide a value and citation for the required bond strength for MEMS packages.

- some wording issues

- "rarely" is often used incorrectly. e.g. "Au rarely adheres to..." -> replace with "Au badly adheres to..."

- P2/3: Very confusing use of the words "the/this/in this study", "previous study" etc. Please make it more clear which refer to new work by the authors, the authors' previous work, or research by others.

- smoothing -> smoothening

- p5, line 117/118: please reformulate. It's not clear what the authors want to say here.

- p11, line 192, and p14 line 253: replace "supposedly" with "presumably"

In conclusion: The manuscript shows good work, but it feels very thin and lacking substance. Had the authors included a sealed cavity, hermeticity testing, and performed a getter activation, then this would be a very nice paper indeed.

Author Response

Comment 1-1

- It's a real shame the authors did not perform any experiments on hermeticity (e.g. bonding to a simple device substrate with cavities and checking the leak rate by residual gas analysis). Without it, this manuscript is only about a bonding step of two flat Au surfaces, which much less impactful.

In it's current form, the manuscript does not prove whether this bonding method is hermetic or not.

Answer 1-1

Thank you for your comments. Although we tried to perform the packaging test, unfortunately, the cavity etching equipment has been broken. We are terribly sorry for the lack of experiments on hermeticity.

Meanwhile, we have reported that the bonding interface with voids of ~10 nm diameter can form hermetic sealing. The bonded cavity structures with a volume of 160 μL were maintained in a vacuum, as shown in the following table (see red square: the test result using the package formed by room temperature bonding). [Reference #26]

In these studies, AFM observation indicates that the RMS surface roughness of the Au surface was 0.5 nm, and the nano-voids were present at the bonding interface (see the right TEM image) [Reference #25]. These AFM and TEM results revealed that the surface and interface structures are similar to those using the transferred getter film.

Thus, we believed that the vacuum packaging can be fabricated using the transferred smooth getter film. Again, we apologize that we cannot perform the packaging test.

 We added the following sentence at P. 13, L242-245.

In our previous studies, the Au/Au bonding interface with such voids could form hermetic sealing at room temperature[25,26]. In addition, the hermeticity was improved by the following annealing step for getter activation[26].

Comment 1-2

- An analysis of the bond interface, bond strength, voids, and preferably also hermeticity after a getter activation annealing is missing. The authors only show these analyses after bonding, without annealing at 200°C (or 300 °C as stated in the authors' previous paper). This is important since a real device packaged this way would spend most of its life after annealing.

Answer 1-2

It is true that the real devices are used after the getter-activation annealing step. The annealing step initiates the thermal diffusion of Au atoms across the bonding and improves the bonding quality. The following TEM image is the Au/Au interface after annealing at 300  °C[13]; the nanovoids at the bonding interface disappeared. As shown in the blue square in the previous table, the annealing process improved the hermeticity, as the amount of gas molecules was reduced.

The present study evaluates the bonding quality before annealing, which was significantly high. It is expected that the interfacial strength and hermeticity would be improved by annealing. However, the problem of voids caused by particles is difficult to be solved by annealing. Thus, the cleanliness of the process room should be improved.

Comment 1-3

- Almost 1/2 of all citations are self-citations, or associated citations. Please include more research from other authors, especially about hermetic metal-metal bonding, and about atomically flat metal surfaces.

Answer 1-3

Thank you for your comment. We added references and 17/26 were the studies in other people.

Comment 1-4

- The authors don't address the issue of how to achieve atomically flat Au surfaces on a real device wafer. How feasible is this method in reality?

Answer 1-4

One possible approach is the use of a cap wafer coated with the smooth getter layer, as shown in Fig. 1.

Another possible approach is to transfer the smooth getter layer around MEMS devices and package them, as illustrated in the following figure.

We added the following sentence at P. 2, L48-49.

In addition, the direct bonding of MEMS device wafers having NEG films with cap wafers also allows a simple vacuum packaging process.

Comment 1-5

- Please explain how exactly the exfoliation process was performed.

Answer 1-5

 We have added the explanation of the exfoliation step as follows in P.3 L 92-93:

After the bonding step, a 100-µm-thick razor blade was inserted between the bonded wafers. This causes the exfoliation of wafers at the Au/SiO2 interface.

Comment 1-6

- Figure 2,5,8,etc: the colour of the SiO2 substrate could be confused with a pure SiO2 material, rather than an oxidised Si wafer. Please indicate the oxidised Si wafer more clearly.

Answer 1-6

We modified Figures 3 and 6 as follows:

In addition, we have added the photograph of the oxidized Si wafers to the supplementary material.

Supplementary material

Figure S1. Thermally oxidized Si wafer having a 300-nm-thick SiO2 layer used as Template wafer.

Comment 1-7

- In several places, "grain growth" or "growth surface" are used to explain the roughness of the exfoliated Au surface. This is very unclear. Does it refer to the deposition process, or the bonding step at 200°C? Please explain this better.

Answer 1-7

Thank you for your comment. In this manuscript, we meant the grain growth in the deposition process. We added the red text as follows:

(P. 9, L. 173-174)

The surface of the Au/Ti/Au layer on the SiO2 template was rough because the RMS roughness was 3.48 ± 0.09 nm owing to grain growth during the deposition step.

(P. 9, L. 177-179)

This is possibly because the surface exfoliated from the template hardly suffered from the increase in the roughness of the thick layer. (P. 9, L. 177-179)

Comment 1-8

- Was the tensile bond strength test performed on both Fig 5 and Fig 8 samples, or only Fig 8?

Answer 1-8

We added the red text as follows:

It was the diced specimen shown in Fig. 8. We modified the texts as follows.

The mechanical strength of the bonded substrates shown in Fig. 8 was evaluated using a tensile tester. The bonded Si substrates were diced into 10-mm-square chips and glued to jigs for the tensile test. (P. 11, L. 205-207)

Comment 1-9

- p11, line 200: Please provide a value and citation for the required bond strength for MEMS packages.

Answer 1-9

In a tensile test, there are no criteria for tensile strength. When a tensile strength over 10 MPa is applied, a fracture sometimes occurs within the Si bulk. Thus, it is believed that the bonding strength in the present study was sufficiently high.

We added the following sentence in P. 11 L. 209-211.

It is known that the interfacial strength over 10 MPa is equivalent to the bulk strength of Si substrates.

Comment 1-9

 - "rarely" is often used incorrectly. e.g. "Au rarely adheres to..." -> replace with "Au badly adheres to..."

Answer 1-9

We modified the texts as follows.

(P.3, L. 85-87)

This is because the oxidized Si substrate has an atomically smooth surface and the Au film badly adheres to the SiO2 surface.

(P.4, L. 113-115)

Because the deposited Au layer badly adhered to the SiO2 surface, the Au/SiO2 interface could be exfoliated without significant mechanical deformation.

(P.9, L. 177-179)

This is possibly because the surface exfoliated from the template hardly suffered from the increase in the roughness of the thick layer.

(P.12, L. 224-226)

Meanwhile, gaps were hardly observed at the second bonding interface because the Au surface after the transfer step was significantly smooth (RMS: 0.538 ± 0.12 nm).

Comment 1-10

 - P2/3: Very confusing use of the words "the/this/in this study", "previous study" etc. Please make it more clear which refer to new work by the authors, the authors' previous work, or research by others.

Answer 1-10

We modified the red text as follows.

(P.2, L. 64)

To address this dilemma, the present study developed fabrication techniques for an Au/Ti-based getter layer with a smooth Au surface and a thick Ti layer.

(P.2, L. 71)

Our previous study demonstrated the TS process of a 2-mm-square Au/Ti-metallized chip and achieved room-temperature bonding using a smooth surface [19].

(P.3, L. 74)

In the present study, we fabricated getter films with a smooth Au surface and a 100-nm-thick Ti layer using two TS-based methods.

Comment 1-11

 - smoothing -> smoothening

Answer 1-11

We modified the texts as follows. (P. 4, L. 106)

3.1. Smoothening Au/Ti getter film by conventional TS technique

Comment 1-12

 - p5, line 117/118: please reformulate. It's not clear what the authors want to say here.

Answer 1-12

We added the sentences as follows. (P. 5, L. 121-124)

The photograph of the thermally-oxidized Si substrate (template) is shown in the supplementary material. The appearance of the bare template substrate and that after the TS step is similar. It indicates that most of the Au film on the template substrate was transferred to the packaging substrate.

Comment 1-13

 - p11, line 192, and p14 line 253: replace "supposedly" with "presumably"

Answer 1-13

We modified the red text as follows.

(P. 11, L. 199-201)

Additionally, other voids were presumably caused by the new particles adhering to the surfaces after the transfer process.

(P. 14, L. 260-261)

Because this process enables direct bonding of the smooth Au surface exfoliated from the template, the quality of the packaging is presumably less affected by the increased roughness of the thick film.

Reviewer 2 Report

In this paper, the getter films with a smooth Au surface and a 100-nm-thick Ti layer using two TS-based methods. The smoothness of the Au surface and bonding formation at room temperature were evaluated for the MEMS packaging process. In MEMS technology, the common bonding technology is Au-Au heat pressure bonding, which can reduce the dependence on metal surface roughness. Please add the necessity and application scenario of room temperature bonding in introduction section.

Author Response

Comment 2-1

In MEMS technology, the common bonding technology is Au-Au heat pressure bonding, which can reduce the dependence on metal surface roughness. Please add the necessity and application scenario of room temperature bonding in introduction section.

Answer 2-1

One of the major advantages of room temperature bonding is that it allows the integration of dissimilar materials regardless of mismatches in the coefficient of thermal expansion (CTE). This merit is meaningful in the MEMS industry because various kinds of material (e.g. Si, Glass, Piezoelectric materials) were used in the microdevices.

In addition, the annealing process before the bonding may cause thermal diffusion of Ti atoms to the surface and deteriorate the bonding of Au surfaces.

We added the following sentences in P.1, L.34-36.

Several types of micro-electro-mechanical systems (MEMS) are degraded by gas molecules in packages. Thus, the vacuum packaging process is an important step in the performance and lifetime of MEMS devices. Low-temperature bonding techniques have been extensively studied for the packaging of MEMS assemblies to suppress thermal damage [1–7]. Particularly, atomically smooth Au surfaces can form atomic bonds at room temperature [8–10], which enables the integration of dissimilar materials regardless of the mismatches in the coefficient of thermal expansion. It has been demonstrated that wafers coated with Au/Ti (from top to bottom) bonding layers can form vacuum packaging at room temperature [11]. In these previous studies [8,9,11] , the Ti layer was deposited to form a strong adhesion between the Au bonding layer and MEMS substrate.

Reviewer 3 Report

In the discussion of previous work (line 53, page 2), it would be helpful to describe how the Au/Ti layers were deposited in that work.

The explanation of the processing (lines 81-83) is a bit confusing and difficult to follow.  Expand the discussion of how the samples are fabricated so the reader can more easily follow.

The explanation of figure 3 is a bit confusing (line 117, page 5)

It would be helpful to provide additional data to back up the claim that unbonded areas are due to new particles (line 192 page 11)  Could SEM or TEM be done in those areas to characterize why they were not bonded?

Author Response

Comment 3-1

In the discussion of previous work (line 53, page 2), it would be helpful to describe how the Au/Ti layers were deposited in that work.

Answer 3-1

We added the following sentences in P.2, L55.

Studies on direct bonding typically utilize thin Au/Ti layers, such as sputter-deposited Au and Ti layers with thicknesses of 15 and 3 nm, respectively [5].

Comment 3-2

The explanation of the processing (lines 81-83) is a bit confusing and difficult to follow.  Expand the discussion of how the samples are fabricated so the reader can more easily follow.

Answer 3-2

We added the following sentences in P. 3, L 83-84.

The metal films were formed on oxidized 4-inch-diameter Si substrates; a 300-nm-thick SiO2 layer was fabricated on the Si substrates in a thermal oxidation furnace.

Comment 3-3

The explanation of figure 3 is a bit confusing (line 117, page 5)

Answer 3-3

We modified the following sentences in P. 5, L. 121-124 and added the photograph of the oxidized Si template wafer to the supplementary material.

The photograph of the thermally-oxidized Si substrate (template) is shown in the supplementary material. The appearance of the bare template substrate and that after the TS step is similar. It indicates that most of the Au film on the template substrate was transferred to the packaging substrate.  

Supplementary material

Figure S1. Thermally oxidized Si wafer having a 300-nm-thick SiO2 layer used as Template wafer.

Comment 3-4

It would be helpful to provide additional data to back up the claim that unbonded areas are due to new particles (line 192 page 11)  Could SEM or TEM be done in those areas to characterize why they were not bonded?

Answer 3-4

It is known that the round shape voids are typical because of the particles. The bonded areas caused by other reasons often have an asymmetrical shape. Unfortunately, it is technically difficult to perform SEM observation for finding the particles at the bonding interface.

Reviewer 4 Report

In this manuscript, authors present a template stripping approach to achieve room-temperature bonding using Ti/Au as a getter layer for the vacuum packaging. Although similar method has been proposed previously, the presented works improve the overall quality of deposited layer and could be very helpful for various purposes. Overall, this work is performed with decent writing and appropriate manner. I think the manuscript could be suitable for publication in Sensors after addressing a few concerns below.

1.     Yield: since this paper focus on the fabrication process improvement. If the wafer is diced into standard size die, according to your Fig. 9, can you calculate the fabrication yield, and discuss how you can improve?

2.     Thickness vs. roughness: it seems that in the first approach, the thickness of Au film that exfoliated from template substrate would affect the resulting roughness. Have you tried different thickness of Au film and see if the RMS can be increase/decreased accordingly?

3.     Ti thickness: similar with the 2nd comment, since this manuscript is aiming to provide a better getter layer for vacuum packaging, a thicker Ti layer would be desired. In this case, I think it would be better for authors to try increase the Ti layer thickness from previously reported 100 nm, and see if the proposed approach could achieve decent RMS result.

4.     Comparing the approach #1 (Fig. 2) and #2 (Fig. 5), when being converted to the fabrication process in Fig. 8, the approach #1 will result a (SiO2/Ti 100nm/Au 6nm/Au 20nm/Au 20nm/Ti 5nm/SiO2) structure. In this case, this multi-sandwich structure also includes only one adjacent Au-Au layer which have the potential roughness issue. Authors need to provide a clarification why the second approach could help alleviated this dilemma.

Author Response

Comment 4-1

Yield: since this paper focus on the fabrication process improvement. If the wafer is diced into standard size die, according to your Fig. 9, can you calculate the fabrication yield, and discuss how you can improve?

Answer 4-1

We calculated the number of bright pixels, and the result indicate that 9.5% of the Au films were not bonded. In the present study, the wafers were handled by ourselves; this presumably cause the main source of particles. We believe that the yield would be improved by the machine handling in a clean room.

We added the following sentences in P. 11, L. 201-203.

The ratio of unbonded areas is 9.5 %; it is expected that these unbonded areas will possibly disappear when the cleanliness of the process environment is improved.

Comment 4-2

Thickness vs. roughness: it seems that in the first approach, the thickness of Au film that exfoliated from template substrate would affect the resulting roughness. Have you tried different thickness of Au film and see if the RMS can be increase/decreased accordingly?

Answer 4-2

We are sorry that the additional experiments using films with different thicknesses are difficult. The relation between “Thickness vs. roughness” has been known as follows. To suppress the roughness increase, the ~10-20-nm-thick Au layer is suitable for the bonding process.

We modified the sentences in P. 2, L. 57-59.

The previous studies also revealed that thick Au/Ti films are undesirable for direct bonding because the surface roughness of Au [11] and Ti [14] layers increases with the increase in the film thickness by the grain growth during deposition.

Comment 4-3

 Ti thickness: similar with the 2nd comment, since this manuscript is aiming to provide a better getter layer for vacuum packaging, a thicker Ti layer would be desired. In this case, I think it would be better for authors to try increase the Ti layer thickness from previously reported 100 nm, and see if the proposed approach could achieve decent RMS result.

Answer 4-3

In the case of the Ti layer, the relation has been known as follows. The Au/Ti/Au layer was terribly rough for the hermetic sealing. However, the exfoliated surface is atomically smooth even when the thickness is 100 nm.

Comment 4-4

Comparing the approach #1 (Fig. 2) and #2 (Fig. 5), when being converted to the fabrication process in Fig. 8, the approach #1 will result a (SiO2/Ti 100nm/Au 6nm/Au 20nm/Au 20nm/Ti 5nm/SiO2) structure. In this case, this multi-sandwich structure also includes only one adjacent Au-Au layer which have the potential roughness issue. Authors need to provide a clarification why the second approach could help alleviated this dilemma.

Answer 4-4

Approach #1(Fig. 2) attempts the smoothening of the rough surface of thick film, as follows. Thus, it is easily affected by the roughness increase by the grain growth in the thick Ti film.

Meanwhile, approach #2(Fig. 5) utilizes the Au surface deposited onto the smooth template. This study revealed that the roughness of the Au surface was hardly affected by the roughness increase in the thick metal film.

We added the following sentences.

(P. 7, L. 151-154)

Unlike the conventional TS process, which smoothens the surface of the thick Au/Ti layer, it is expected that the bonding surface can have a morphology similar to that of the smooth template and is less affected by the increased roughness of the thick layers.

(P. 9, L. 178-180)

This is possibly because the surface exfoliated from the template hardly suffered from the increase in the roughness of the thick layer.

(P. 14, L. 261-263)

Because the second approach enables the exfoliation of the atomically smooth Au surface from the template, the quality of the packaging is presumably less affected by the increased roughness of the thick film

Round 2

Reviewer 4 Report

Comments are well addressed. Good for publication in present form.